# Nanomechanical Properties of Articular Cartilage Due to the PRP Injection in Experimental Osteoarthritis in Rabbits

**DOI:** 10.3390/molecules25163734

**Published:** 2020-08-15

**Authors:** Mikhail Ihnatouski, Jolanta Pauk, Boris Karev, Dmitrij Karev

**Affiliations:** 1Scientific and Research Department, Yanka Kupala State University of Grodno, Grodno, Ozheshko str., 22, 230023 Grodno, Belarus; mii_by@mail.ru; 2Mechanical Engineering Department, Bialystok University of Technology, Biomedical Engineering Institute, Wiejska 45A, 15-351 Bialystok, Poland; 3Department of Orthopedic and Traumatology, Grodno City Emergency Hospital, Sovietskih Pogranichnikov str., 115, 230027 Grodno, Belarus; bkarev@gmail.com; 4Department of Traumatology, Orthopedics and Field Surgery, Grodno State Medical University, Gorkogo str. 80, 230009 Grodno, Belarus; dmitriy.karev@gmail.com

**Keywords:** hyaline cartilage, platelet-rich plasma therapy, induced model, osteoarthritis, atomic force microscopy

## Abstract

The purpose of this study was twofold. Firstly, we proposed a measurement protocol for the atomic force microscopy (AFM) method to determine the nanomechanical properties of articular cartilage in experimental osteoarthritis in rabbits. Then, we verified if mechanical properties can be evaluated with AFM shortly after platelet-rich plasma (PRP) injection. We hypothesized that the modulus determined by AFM indentation experiments could be utilized as a progressive disease marker during the treatment of osteoarthritis. The rabbits were equally divided into three groups of six: control (group 1); injections of saline (0.5 mL) and 10% surgical talc (Talcum Pharmaceutical^®^, Minsk, Belarus) were delivered into the right knee under the patella (group 2 and 3); and PRP was injected into the right knee (group 3). In group 2, the arithmetic average of absolute values (Ra) change was a 25% increase; the maximum peak height (Rp) increased by over 102%, while the mean spacing between local peaks (S) increased by 28% (*p* < 0.05). In group 3, Ra increased by 14% and Rp increased by 32%, while S decreased by 75% (*p* < 0.05). The Young’s modulus of the surface layers decreased by 18% as a result of induced model of osteoarthritis (IMO) (*p* < 0.05), and it increased by 9% (*p* < 0.05) as a result of PRP therapy, which means that the mechanical properties of cartilage were partially recovered. This research demonstrates that Young’s modulus utilized on a nanometer scale has potential to be a progressive disease marker during the treatment of osteoarthritis.

## 1. Introduction

Osteoarthritis (OA) is a musculoskeletal disease common throughout the world, affecting 10–15% of the population aged 45 years and older [1,2]; over 30 million people in the United States [3] and 302 million worldwide suffer from OA [4]. Currently, the disease ranks 12th globally among all the causes of disability in the population [4]. The prevalence of osteoarthritis is increasing, and it is expected that 78 million adults in the US will have been affected by 2040 [5]. It is a chronic progressive disease whose causes include imbalances between anabolic and catabolic processes in joint tissues [6]. The risk factors include old age, obesity, and type 2 diabetes mellitus, previous damage, or mechanical overload to the joint [7]. Hyaline cartilage undergoes degenerative-dystrophic changes with subsequent involvement of the subchondral bone [8,9]. Moreover, chronic synovitis, sclerosis of the joint capsule, meniscal degeneration, and articular and muscular atrophy are also observed. Osteoarthritis is indicated as one of the most common forms of arthritis, with the knee joint being the most commonly affected site [6,7,10,11,12].

Preclinically, the pathogenesis, pathophysiology, and the effect of therapeutic intervention in OA treatment require the development of animal models, which play a crucial role in the understanding of the disease. Such models are classified in literature based on their etiology, i.e., as primary osteoarthritis, post-traumatic osteoarthritis, etc. [12]. Both small (mouse, rat, rabbit, and guinea pig) and large animals (dog, goat/sheep, and horse) have been used to develop OA models [12,13,14,15,16]. Chemically induced models are mainly used for the assessment of the effectiveness of treatment in inflammation. They involve the injection of an inflammatory compound (sodium monoiodoacetate, quinolone, collagenase, etc.) directly into the knee joint to induce OA in animals [12]. Studies of histopathological outcomes have been widely published in [15]; however, the mechanical properties and the submicron surface morphology of animal hyaline cartilage remain unclear.

In the 21st century, orthopedists’ attention turned to conservative methods of joint disease treatment, indicating that new modalities should focus on multifunctional treatment and solving a set of biotribological problems [16,17]. Several treatment options were used to stop the progression of inflammation. These include conservative methods, non-steroidal anti-inflammatory drugs, glucocorticosteroids, and hyaluronic acid, as well as therapeutic exercise and assistive devices [16]. Current studies involve the use of platelet-rich plasma therapy (PRP) in the treatment of osteoarthritis because it is easy to prepare and well-tolerated. Some basic, preclinical, and clinical case studies report the ability of platelet-rich plasma to improve musculoskeletal conditions, including osteoarthritis [12,16,18,19,20,21,22,23,24]. Nevertheless, there are still concerns regarding its clinical efficiency, mainly due to the heterogeneity of the used methods and the obtained results, lack of standardization, and quantification of the PRP protocols [25,26,27,28,29,30,31]. Defining additional parameters related to articular cartilage properties is desirable.

Despite the numerous advantages of available imaging modalities, their limitations are in ultrastructural details at molecular resolution, the complexity of sample preparation procedures, and the lack of the morphological distortion of cells at nanometer resolution [32,33,34,35]. Osteoarthritis appears as a result of changes in elasticity of cells and tissues; hence, it is crucial to consider aspects such as the mechanical properties and the submicron surface morphology of hyaline cartilage. Lately, atomic force microscopy was recognized as a powerful tool to assess the properties of articular cartilage [9,36,37,38,39,40,41,42,43,44]. Therefore, the purpose of this study was twofold. Firstly, we proposed a measurement protocol for the AFM method to determine the nanomechanical properties of articular cartilage in experimental osteoarthritis in rabbits. Then, we verified if mechanical properties can be evaluated with AFM shortly after PRP injection. We hypothesized that modulus determined by AFM indentation experiments could be utilized as a progressive disease marker during the treatment of osteoarthritis.

## 2. Results

Within several minutes following the injection, all the animals were conscious and started to move. None of the rabbits showed any sign of a knee infection or swelling or died during the test. From the first day, rabbits developed synovitis, manifested as an increase in the local temperature and the diameter of the joint. Two independent surgeons, without prior knowledge of the experimental groups, performed the observation.

### 2.1. Histologic Findings according to the Non-Treated and PRP-Injected Knees

The difference of histologic scores in group 3 was higher (3.3 (1.4)) than those in group 2 (2.1 (1.8)), *p* < 0.05. Histologic examination showed nearly normal cartilage surfaces with normal cellularity in the transitional and radial zones in group 3 with PRP injection.

### 2.2. Submicron Surface Morphology of Rabbits Cartilage and the Mechanical Properties of Rabbits

AFM images in the three groups were conducted in contact mode in the three groups (Figure 1).

Damage and gaps were found on the cartilage tissue in group 2. Although the destruction sites were also found on cartilage in group 3, the size of the lesions decreased after PRP therapy. The roughness parameters of cartilage are presented in Figure 2.

In group 2, the maximum of the arithmetic average of absolute values (Ra) change was a 25% increase; the maximum peak height (Rp) increased by 102%, while the mean spacing between local peaks (S) increased by 28%, compared to the control group (*p* < 0.05). Thus, the wear caused by artificial osteoarthritis differs from the submicron surface morphology that appears during the natural genesis and course of osteoarthritis which was described in [44]. This type of wear can be explained by mechanical damage caused by injections of surgical talc and the short duration of illness. In group 3, Ra increased by 14%; Rp increased by 32%, while S decreased by 75%, compared to the control group (*p* < 0.05). The same values amounted to an Ra decrease of 10%, Rp decrease of 41%, and S decrease of 40%, compared to group 2 (*p* < 0.05). We found that the restoration of the surface morphology as a result of therapy was recorded. This is because the altitude parameters decreased below the initial value, as did the mean spacing between local peaks, which can be regarded as an increase in the density of small surface irregularities and the smoothing out of large ones. Our study revealed a significant decrease in the Young’s modulus of the articular cartilage associated with OA disease. The corresponding indentation depth was 25–150 nm (Table 1).

The results have shown that the Young’s modulus of a healthy joint cartilage ranges from 1.95 MPa on the surface to 0.64 MPa at a depth of 150 nm. The range of the Young’s modulus of the rabbits’ cartilage in group 2 was from 1.60 to 0.70 MPa; the range of the Young’s modulus in group 3 was from 1.71 to 0.71 MPa. Thus, it was found that the mechanical properties of hyaline cartilage deteriorate under the influence of simulated osteoarthritis; the Young’s modulus of the surface layers decreased by 18% as a result of IMO (*p* < 0.05), and it increased by 9% (*p* < 0.05) as a result of PRP therapy, which means that the mechanical properties of cartilage were partially recovered.

## 3. Discussion

Although animal models do not reflect all aspects of human pathology, they are still used in preclinical trials [12,14,23,45,46]. They play a significant role in understanding OA and the effects of therapeutic interventions. An induced model causing the initial stages of primary osteoarthritis in rabbits was used in this study. There are some differences between human and rabbit. Human weight-bearing is primarily locked on the knee extension, while rabbit hind limbs are usually kept in a fully flexed position [46]. Additionally, the cell densities are 1800 and 7500 per mm in humans and rabbits, respectively. Moreover, rabbit stifle joints have different load characteristics and cartilage thickness compared to humans [46]. Despite those differences, rabbits provide many advantages as they are easy to handle and to house and are cost-effective. Additionally, the rabbit model is the most adequate for this study because it is suitable for the assessment of cartilage repair; rabbits have large enough joints and appropriate size for simple surgical procedures and handling of specimens with a cartilage thickness of 0.25–0.75 mm [47]. The bone mineral density (1.19 g/cm^3^) is similar to humans at the bone plate (1.17 g/cm^3^) [23]. Rabbits were used for the assessment of cartilage repair in studies lasting up to 16 weeks [48,49], and animal models of OA in which articular cartilage damage was induced with intra-articular collagenase injection have proved to be similar to human OA [50].

There is a significant gap regarding PRP clinical efficiency. Development of new methods and quantification of the PRP protocols are still under discussion. There are studies in the literature that tested the effect of PRP when used to treat OA in a rabbit model. In [51,52], the authors proved that PRP injection is regenerative in the collagenase-induced knee OA. The PRP treatment group demonstrated a greater extent of cartilage regeneration, as well as higher production of glycosaminoglycans in the extracellular matrix. In [53], the authors reported that the PRP-treated rabbits had the progression of osteoarthritis significantly decreased. Serra et al. [54] also used a rabbit model analyzing the effects of PRP on full-thickness articular cartilage defects. They found that the control group was both microscopically and macroscopically superior concerning defect filling, whereas the PRP group did not show better results compared with the placebo. Our study reveals that the difference of histologic scores between the non-treated group and PRP injected knees in group 3 was higher than those in group 2, which suggests that PRP injection is regenerative. In another research study, the OA changes were induced by different doses of 0.5–2.0 mg by the intra-articular injection of [24,51,55,56,57]. In our study, the dose was 0.5 mg and frequency of PRP injection was determined as four doses at seven-day intervals. It is in agreement with [57], in which the authors stated that the therapeutic effects of plasma therapy are based on the release from platelets-flat, nuclear-free cells that circulate in the bloodstream with a lifespan of 7–10 days.

Radiography, currently the gold standard for OA imaging, and other image modalities have limitations in the area of providing information about the mechanical properties of cartilage altered by degenerative joint disease. The atomic force microscopy (AFM) was widely utilized in the detection of pathological and the biomechanical properties of biological target structures on microscopic scale conditions [58,59,60,61,62,63,64]. In our study, AFM was used to test the mechanical properties and the submicron surface morphology of rabbits’ hyaline cartilage. Pronounced heterogeneity of the biological surfaces’ topography renders the utilization of AFM method quite challenging. Thus, we proposed a measurement protocol for the AFM method to determine the nanomechanical properties of articular cartilage. We pre-selected the silicon probes with a single radius of the needle point using a scanning electron microscope (±1 nm) and with cantilever constant (±0.01 N/m) by testing samples with the previously known Young’s modulus (NaCl crystal with *E* = 37 GPa) to reduce error. The AFM indentation mode can be realized in a few ways depending on the set controlled parameter. We used the set-point as a controlled parameter. The set-point corresponds to relative deflection of the probe cantilever in the static mode of indentation. AFM moves the sample step-by-step towards the probe in such way that set-point changes in the preset range with equal steps between the set-point values. Measured cantilever deflection is a function of set-point. The dependence of the cantilever deviation on the set-point parameter is established, based on the measurement results. The cantilever deflection can be directly converted into force if the mechanical properties of the cantilever are known. Chandran et al. [65] stated that the indentation depth of at least 0.6 μm is required to obtain values of matrix elasticity. It was observed that each 60 μm change in indenter location could result in a twentyfold variation of the measurement [66]. Our previous research carried out to 450 nm in depth [67] shows that the surface is more suitable to diagnose, so in this study, we measured the mechanical properties of cartilage surface to 150 nm in depth. However, the statistical significance of the results was observed below 100 nm. Past studies of articular cartilage have assumed a wide range of Poisson’s ratios [63,64]. In this study [67], the local Poisson’s ratio was assumed to be 0.5 for articular cartilage based on previous experimental reports. The changes in articular cartilage structure result in deterioration of the mechanical strength [60]. The roughness parameters of hyaline cartilage surface: the absolute value (Ra), the maximum peak height (Rp), and the mean spacing between local peaks (S) were measured for three groups: control; induced model of osteoarthritis; and the group with PRP treatment. Our results show that PRP injection into the knee under the patella can improve the mechanical properties and the submicron surface morphology of rabbits’ hyaline cartilage: Ra and Rp increased, while S decreased compared to healthy rabbits’ cartilage (*p* < 0.05). Moreover, Ra, Rp, and S decreased compared to group 2. The extent of mechanical compliance can be expressed as Young’s modulus, which describes the elastic properties of a material [61,62]. In our study, the indentation clearly revealed a decrease in the Young’s modulus of the upper layers of hyaline cartilage (up to 100 nm). It increased by 9% as a result of PRP therapy, which means that the mechanical properties of cartilage partially recovered. Usually, the treatment effects require monitoring at longer time period. Therefore, we speculate that 0.5 mg PRP injection is sufficient to evaluate morphological changes with AFM shortly after PRP injection.

Although the roughness measurements are routine in the presented field, the novelty of our study lies in using surface nano-roughness and microhardness of subsurface layers in describing changes in articular cartilage properties during development of OA. These parameters can be treated as essential indicators of the pathologic progression of OA diseases. In addition, the elastic modulus determined by AFM indentation experiments, can be utilized as a progressive disease marker in preventing the development of OA.

To the best of the authors’ knowledge, this is the first study to assess the response to treatment consisting of intra-articular PRP injection on the mechanical properties and the submicron surface morphology of hyaline cartilage. The limitations of this study included the sample size and lack of therapeutic effects of PRP in severe knee OA over a long time. In the future, we will consider using a liquid cell for AFM measurements, because it eliminates surface tension and makes it possible to perform measurements immediately after obtaining the specimens.

## 4. Methods

### 4.1. Induced Rabbits Model of Osteoarthritis

Eighteen rabbits (chinchilla) from the Faculty of Veterinary Medicine at Grodno State Agrarian University, aged five months and weighing from 2400 to 3500 g, were included in the study. The rabbits were equally divided into three groups of six: control (group 1); injections of saline (0.5 mL) and 10% surgical talc (Talcum Pharmaceutical^®^, Minsk, Belarus) were delivered into the right knee under the patella (group 2 and 3); and PRP was injected into the right knee (group 3). The inclusion criteria were good appetite and activity, a shiny even coat, a clean nose and eyes, a body temperature in the range 38.5–39.5 Celsius degrees, a relative humidity of 49 ± 8%, a heart rate ranging from 120 to 160 BPM, and a respiratory rate ranging from 50 to 60 movements per 120 min. The exclusion criteria included lack of appetite, intestinal disease, inactivity, or suppressed behavior. Experiments on animals were performed in accordance with internationally accredited guidelines, and had been approved by the Ministry of Health of the Republic of Belarus (No. 3, 13 January 2017). The rabbits were anesthetized by intramuscular administration of 10 mg/kg of xylazine (Ksilanit^®^, Saratov, Russia) and 50 mg/kg of ketamine (Ketamine^®^, Moscow, Russia) which kept the animal sedated for a longer time with minimal pain. Xylazine coadministered with ketamine is a safe anesthetic adjunct to induce short periods of surgical anesthesia. In [40], it was proved that the combination of both xylazine and ketamine cause rabbits’ muscle relaxation; rabbits have a smoother emergence from anesthesia.

### 4.2. Platelet-Rich Plasma (PRP) Preparation

For PRP preparation, 9.0 mL of venous blood were drawn from the marginal auricular vein of the rabbit’s ear and mixed with 0.5 mL of 0.0775 mol/L sodium citrate acting as an anticoagulant. The blood was centrifuged at 1200 rpm (160 g) for 20 min to separate the plasma containing platelets from red cells. The plasma was drawn from the top layer and centrifuged for an additional 15 min at 2000 rpm (450 g) to separate the platelets at a temperature of +18 to +22 Celsius degrees. After centrifugation, 0.6–0.7 mL of PRP was drawn from a plasma layer rich in platelets using a syringe and an injection needle [39]. Six rabbits underwent platelet-rich plasma therapy (group 3), while the other six did not receive PRP treatment (group 2). 0.5 mL of PRP were injected into right knee, and the injection dose was repeated in seven-day intervals. Groups 2 and 3 were quarantined for 14 days. Then, group 2 was euthanized by CO inhalation, and group 3 started PRP therapy. The first PRP injection was performed directly after surgery.

### 4.3. Histologic Parameters

The rabbit’s articular cartilage was harvested after euthanasia from the femoral condyles, yielding ∼5 mm × 5 mm pieces, ∼2 mm thick. All samples were fixed in 10% neutral buffered formalin for at least 48 h. For histopathological examination, tissue samples were washed with 10% phosphate buffered saline solution, dehydrated in alcohol, and embedded in paraffin wax [13]. The degree of cartilage degradation was assessed using the scoring system modified by Mankin et al. [44]. A pathologist evaluated the structural change of articular cartilage as histological evidence of cartilage degeneration.

### 4.4. Measurement Protocol for AFM Mechanical Properties of Rabbits Cartilage

Atomic force microscopy AFM NT-206 (©MicroTestMachines, Gomel, Belarus) was performed to test submicron surface morphology and to measure the mechanical properties of rabbits’ cartilage specimens in the static scanning mode in air. A CSC38 MikroMasch^®^ silicon probe (MikroMasch, Watsonville, CA, USA) was used. The resulting tip radius was less than 35 nm, whereas the total tip height was 12 to 18 μm. The full tip cone angle was 40 degrees, and the probe material was n-type silicon. A type A cantilever with resonance frequency ranged from 8 to 32 kHz was used. The force constant was from 0.01 to 0.36 N/m; length 250 ± 5 μm; width 32.5 ± 3 μm; thickness 1.0 ± 0.5 μm [44]. The scanning results were assessed with a scan area of Ar = 9 × 9 μm^2^. The SurfaceExplorer (©MicroTestMachines, ver 1.1.5, Gomel, Belarus) software and NanoImages (ver 6.128.15, Mikhail Ihnatouski, Grodno, Belarus) were used to both visualize the experimental data and measure the roughness parameters: the arithmetic average of absolute values (Ra); the maximum peak height (Rp); and the mean spacing between local peaks (S). The parameters were measured at five points of each specimens (15 specimens per rabbit; five measurements per specimen; 1350 measurements in total).

The equations for calculation of Young’s modulus from cantilever deflection data was derived using a linear model of interacting cantilever spring and elastic surface in conditions for quasi-static equilibrium presented as equality of cantilever spring forces exerted and elastic surface response [67]:(1)$$k\cdot Z_{defl}=P\left(h\right)$$
where *k* is stiffness of the cantilever, and *P(h)* is normal load as a function of variable indentation depth. The penetration of the probe *h* into articular cartilage is presented by Equation (2) and in Figure 3 [67]:(2)$$h=Z_{pos}-Z_{defl}$$
where *h* is the penetration of the probe into cartilage, $\left(Z_{defl}\right)$ is the bend of the console, and $\left(Z_{pos}\right)$ is the displacement of the console along the vertical axis.

The Young’s modulus from the depth of penetration into the surface (E(h)) was investigated as follows:(3)$$E\left(h\right)=\frac{3}{4}\left(1-\nu^{2}\right)\frac{k}{\sqrt{R}}\frac{z_{defl}}{h^{3/2}}$$
where *v* = 0.5 is the Poisson’s ratio of the cartilage, *k* = 0.08 N/m is the stiffness of the CSC 38 cantilever, and *R* = 30 nm is the radius of the needle point of the CSC 38.

### 4.5. Statistical Analysis

Statistical analyses were performed using the Statistica software (StatSoft 13.1, Cracow, Poland). The data concerning the roughness parameters and Young’s modulus were checked for normality using the Kolmogorov–Smirnov test. The Kruskal–Wallis test was used to compare the difference of the roughness parameters and Young’s modulus among the three groups as well to compare the difference of histologic scores between non-treated and PRP-injected knees. A *p*-value < 0.05 was considered as statistically significant. All data are presented as a mean ± standard deviation (std).

## 5. Conclusions

The AFM method enables measuring the mechanical properties of hyaline cartilage during the disease process and after applied therapy in the short term. The results reflect the direction and magnitude of cartilage properties changes and show that PRP injection has positive effect on cartilage regeneration in the rabbit’s knee. This research demonstrates that Young’s modulus utilized on a nanometer scale has potential to be a progressive disease marker during the treatment of osteoarthritis.

## Figures and Tables

**Figure 1 molecules-25-03734-f001:**
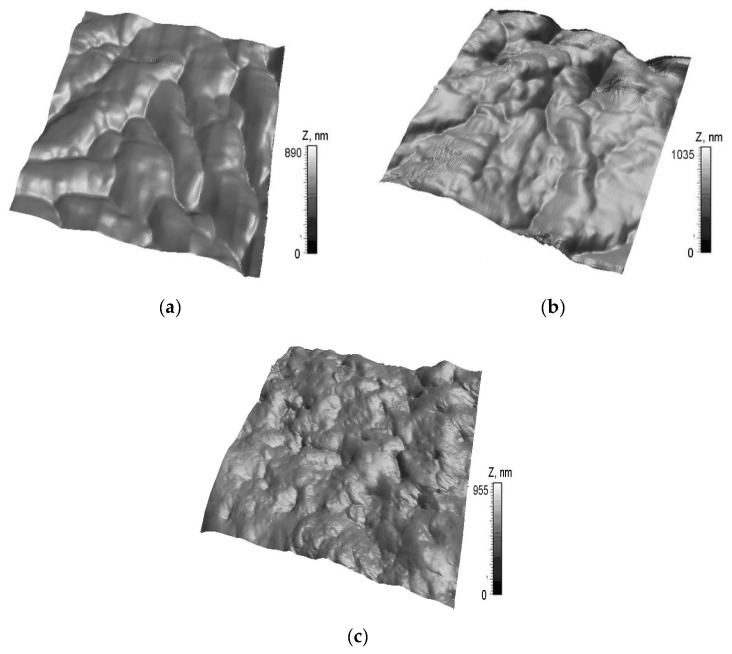
AFM images of rabbit cartilage surfaces 9 × 9 µm^2^: (**a**) group 1; (**b**) group 2; and (**c**) group 3.

**Figure 2 molecules-25-03734-f002:**
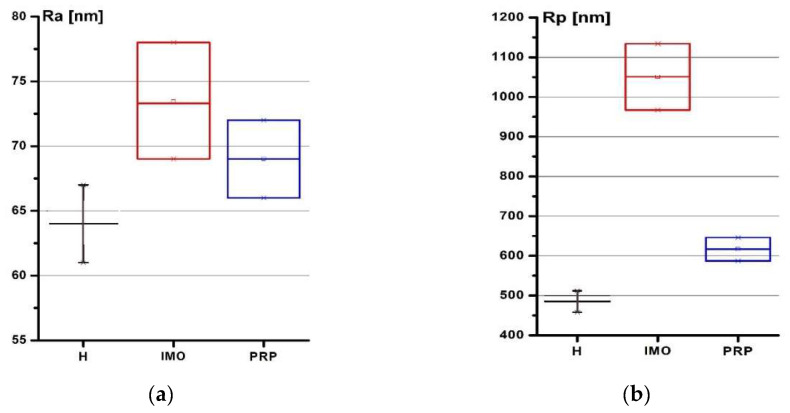
The roughness parameters in group 1 (H), group 2 (IMO—induced model of osteoarthritis), and group 3 (PRP): (**a**) the arithmetic average of absolute values (Ra); (**b**) the maximum peak height (Rp); (**c**) the mean spacing between local peaks (S).

**Figure 3 molecules-25-03734-f003:**
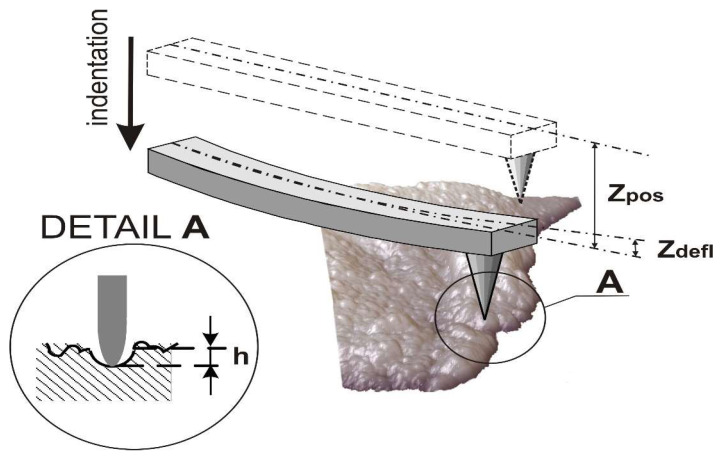
AFM indentation: $Z_{defl}—$the bend of the console;$\;Z_{pos}—$the displacement of the console along the vertical axis; *h*—the penetration of the probe into cartilage.

**Table 1 molecules-25-03734-t001:** The Young’s modulus from the indentation depth.

*h*, nm	Group 1, MPa (std)	Group 2, MPa (std)	Group 3, MPa (std)
25	1.95 (0.035)	1.60 (0.027) *	1.71 (0.022) *
50	1.88 (0.046)	1.53 (0.047) *	1.62 (0.038) *
75	1.37 (0.046)	1.40 (0.050) *	1.50 (0.070) *
100	0.94 (0.020)	1.04 (0.183) **	1.22 (0.075) **
125	0.75 (0.035)	0.79 (0.131) **	0.87 (0.029) **
150	0.64 (0.029)	0.70 (0.082) **	0.71 (0.024) **

* *p* < 0.05, ** *p* < 0.1.

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
