# Peer review of "Nanomechanical Properties of Articular Cartilage Due to the PRP Injection in Experimental Osteoarthritis in Rabbits"

_molecules, 2020, doi:10.3390/molecules25163734_

Round 1
Reviewer 1 Report
The following are concerns that merit a response. Changes where called for should be incorporated into a revised manuscript.
1) Abstract. Line 19, Rabbits should not be described as "patients." The abbreviations Ra, Rp, S and IMO should be defined. OA was not "cured (line 22). Find a better word.
2) Lines 36-38. This information should be referenced.
3) What was the sex of the rabbits? Page 2. Why are there unequal number of rabbits in the groups (6 versus 12)?
4) There was no mention on page 3 regarding how long it took for OA to develop. Was there any sign of inflammation? Any histology performed? If not, why?
5) Section 2.4. Despite the findings that appear to show statistical significance between PRP-treated OA rabbits and the control group a Power Analysis should generally be employed to limit the number of animals needed to show statistical significance. Was a Power Analysis performed? lines 152-153. How were the size of the lesions measured?
7) Reference 13 refers to guinea pigs in the title.
8) Page 7. Was the term "worn out" (line 214) inferred from the measurements? Line 227, a reference is needed.
9. Reference 26. Is the journal Osteoarthrosis and Cartilage or Osteoarthritis and Cartilage (O&C)? Also the volume and page number 2, 1, 8-9 looks strange for a paper in O&C published in 2017.
Author Response
The reviewer has presented us with important comments and requests, which we believe have improved the quality of our manuscript significantly. Thank you very much. In the following pages, we address each and every comment in detail. Each comment is marked by the reviewer’s original text indexed for convenience of reference, and explains what and how changes were implemented in the new submission. Also, we highlighted all changes in the manuscript in red. We hope the revised manuscript is much closer to the high standard of Molecules.
1. Line 19, Rabbits should not be described as "patients." The abbreviations Ra, Rp, S and IMO should be defined. OA was not "cured (line 22). Find a better word.
A: Done.
The purpose of this study was twofold. Firstly, we proposed a measurement protocol for the AFM method to determine the nanomechanical properties of articular cartilage in experimental osteoarthritis in rabbits. Then we verified if mechanical properties can be evaluated with AFM shortly after PRP injection. We hypothesized that modulus determined by AFM indentation experiments could be utilized as a progressive disease marker during the treatment of osteoarthritis. The rabbits were equally divided into three groups of six: control (group 1); injections of saline (0.5 ml) and 10% surgical talc (Talcum Pharmaceutical®, Belarus) were delivered into the right knee under the patella (group 2 and 3); PRP was injected into the right knee (group 3). In the group 2, the arithmetic average of absolute values (Ra) change was a 25% increase; the maximum peak height (Rp) increased by over 102%, while the mean spacing between local peaks (S) increased by 28% (p<0.05). In group 3, Ra increased by 14%, Rp increased by 32%, while S decreased by 75% (p<0.05). The Young’s modulus of the surface layers decreased by 18% as a result of IMO (p<0.05), and it increased by 9% (p<0.05) as a result of PRP therapy, which means that the mechanical properties of cartilage were partially recovered. This research demonstrates that Young’s modulus utilized on a nanometre scale has potential to be a progressive disease marker during the treatment of osteoarthritis.
2. Lines 36-38. This information should be referenced.
A: Done. References [7].
3. What was the sex of the rabbits? Page 2. Why are there unequal number of rabbits in the groups (6 versus 12)?
A: The rabbits were equally divided into three groups of six: control (group 1); injections of saline (0.5 ml) and 10% surgical talc (Talcum Pharmaceutical®, Belarus) were delivered into the right knee under the patella (group 2 and 3); PRP was injected into the right knee (group 3). There was an equal number of both sexes in each group.
4. There was no mention on page 3 regarding how long it took for OA to develop. Was there any sign of inflammation? Any histology performed? If not, why?
A: Within several minutes following the injection, all the animals were conscious and started to move. None of the rabbits showed any sign of a knee infection or swelling or died during the test. From the first day, rabbits developed synovitis, manifested as an increase in the local temperature and the diameter of the joint. Two independent surgeons, without prior knowledge of the experimental groups, performed the observation. We also added:
2.3. Histologic parameters
Rabbits articular cartilage was harvested after euthanasia from the femoral condyles, yielding ∼5 mm × 5 mm pieces, ∼2 mm thick. All samples were fixed in 10% neutral buffered formalin for at least 48 h. For histopathological examination, tissue samples were washed with 10% phosphate buffered saline solution, dehydrated in alcohol and embedded in paraffin wax [13]. The degree of cartilage degradation was assessed using the scoring system modified by Mankin et al. [48]. A pathologist evaluated the structural change of articular cartilage as histological evidence of cartilage degeneration.
3.1. Histologic findings according to the non-treated and PRP-injected knees
The difference of histologic scores in group 3 was higher (3.3 (1.4)) than those in group 2 (2.1 (1.8)), p<0.05. Histologic examination showed nearly normal cartilage surfaces with normal cellularity in the transitional and radial zones in group 3 with PRP injection.
5. Section 2.4. Despite the findings that appear to show statistical significance between PRP-treated OA rabbits and the control group a Power Analysis should generally be employed to limit the number of animals needed to show statistical significance. Was a Power Analysis performed? lines 152-153. How were the size of the lesions measured?
A: The parameters were measured at 5 points of each specimen (15 specimens per rabbit; 5 measurements per specimen; 1350 measurements in total). We had 450 specimens in each group. The lesions have been measured as a maximum peak height.
6. Reference 13 refers to guinea pigs in the title.
A: Corrected. It should be [12].
7. Page 7. Was the term "worn out" (line 214) inferred from the measurements? Line 227, a reference is needed.
We use the names group 1, group 2 and group 3. The sentence was deleted.
8. Reference 26. Is the journal Osteoarthrosis and Cartilage or Osteoarthritis and Cartilage (O&C)? Also the volume and page number 2, 1, 8-9 looks strange for a paper in O&C published in 2017.
A: Corrected. Nieminen, M.T.; Casula, V.; Nevalainen, M.T.; Saarakkala, S. Osteoarthritis year in review 2018: imaging. Osteoathrosis and cartilage 2019, 27, 3, 401-411. doi: 10.1016/j.joca.2018.12.009.

Reviewer 2 Report
In this work the authors described the use of an AFM-based method for determining the morphology and the nanomechanical properties of rabbit osteoarthritic articular cartilages. The results were compared with the ones obtained on healthy cartilage and PRP treated osteoarthritic cartilages.
The study is scientifically sound and the manuscript is well written. Unfortunately, the general aspects presented by the paper (especially those related to the well-known effect of PRP in the treatment of osteoarthrosis) are not very novel in term of overall literature analysis. Furthermore the ATM methodologies have been extensively described as a method for the characterization of hyaline cartilage in different pathological conditions as well as after a number of treatment approaches including PRP injections.
The reviewer would suggest also some minor changes:
Line 40: a reference should be added at the end of the sentence.
Line 66: please add an example(s) about the open questions that are still under investigation regarding the mechanical proprieties of PRP treated osteoarthritic cartilage.
Line 151: please specify what IMO means
Lines 156-159: that part should be mentioned in the material and methods section
Some typo to fix
Author Response
The reviewer has presented us with important comments and requests, which we believe have improved the quality of our manuscript significantly. Thank you very much. In the following pages, we address each and every comment in detail. Each comment is marked by the reviewer’s original text indexed for convenience of reference, and explains what and how changes were implemented in the new submission. Also, we highlighted all changes in the manuscript in red. We hope the revised manuscript is much closer to the high standard of Molecules.
- Line 40: a reference should be added at the end of the sentence.
A: Done, references [8,9].
- Line 66: please add an example(s) about the open questions that are still under investigation regarding the mechanical proprieties of PRP treated osteoarthritic cartilage.
A: In the 21st century, orthopaedists’ attention turned to conservative methods of joint disease treatment, indicating that new modalities should focus on multifunctional treatment and solving a set of biotribological [17,18]. Several treatment options were used to stop the progression of inflammation. These include conservative methods, non-steroidal anti-inflammatory drugs, glucocorticosteroids, hyaluronic acid, as well as therapeutic exercise, and assistive devices [18]. Current studies involve the use of platelet-rich plasma therapy (PRP) in the treatment of osteoarthritis because it is easy to prepare and well-tolerated. Some basic, preclinical, and clinical case studies report the ability of platelet-rich plasma to improve musculoskeletal conditions, including osteoarthritis [12,18-25]. Nevertheless, there are still concerns regarding its clinical efficiency, mainly due to the heterogeneity of the used methods and the obtained results, lack of standardization and quantification of the PRP protocols [27-34]. Defining additional parameters related to articular cartilage properties is desirable.
Despite the numerous advantages of available imaging modalities, their limitation is in ultrastructural details at molecular resolution, the complexity of sample preparation procedures, and lack of the morphological distortion of cells at nanometer resolution [35-38]. Osteoarthritis appears as a result of changes in elasticity of cells and tissues; hence, it is crucial to consider aspects such as the mechanical properties and the submicron surface morphology of hyaline cartilage. Lately, atomic force microscopy was recognized as a powerful tool to assess the properties of articular cartilage [39-48]. Therefore, the purpose of this study was twofold. Firstly, we proposed a measurement protocol for the AFM method to determine the nanomechanical properties of articular cartilage in experimental osteoarthritis in rabbits. Then we verified if mechanical properties can be evaluated with AFM shortly after PRP injection. We hypothesized that modulus determined by AFM indentation experiments could be utilized as a progressive disease marker during the treatment of osteoarthritis.
- Line 151: please specify what IMO means
A: Done. IMO – Induced Model of Osteoarthrosis. It was explained in Fig.3.
- Lines 156-159: that part should be mentioned in the material and methods section.
A: Done.

Reviewer 3 Report
This study reports the use of AFM to measure nanomechanical properties of articular cartilage in rabbits, after surgical induction of osteoarthritis (OA) with or without treatment with PRP injection. The novelty and study design need further justification to give value to the study. More detailed comments are included below.
- Since the study focuses on AFM-based evaluation of articular cartilage, the similarities and differences of rabbit compared to human articular cartilage (nanomechanical properties and other e.g. structural) should be described in more detail. It is stated in the discussion that “Moreover, the size of chondrocytes in human and rabbits articular cartilage do not differ significantly from each other [13].” However, [13] reports a guinea pig model of OA. The reference should reflect a rabbit model of OA.
- A brief description should be included in the introduction on possible mechanisms of action for the beneficial characteristics of PRP in treating OA, as well as the current clinical evidence on their efficacy.
- How was the dose and frequency of PRP injection in rabbits determined? 4 doses at 7-day intervals is a high frequency of injection that is not clinically viable. Please justify the injection protocol used.
- Why was 28 days chosen as the final time point? Usually the treatment effects would require monitoring at longer time points. Also, was the first PRP injection performed directly after the OA surgery? If so, this study is not really looking at the effect of PRP on ‘treating’ OA, but rather on preventing the development of OA after an injury stimulus. This point needs to be raised and addressed in the discussion, specifically on the physiological relevance of the study design.
- There are other studies in the literature that tested the effect of PRP when used to treat OA in a rabbit model, some of which included AFM as one of the outcome measures. These studies should be cited in the discussion and the results of the current study should be compared. Although mentioned in the discussion, the novelty of the current study should be more clearly stated in the context of the existing literature.
- The clinical relevance of this study should be better defined. The main limitation of this study is that only AFM characterisation of the cartilage has been performed, which does not give much useful information in the context of the existing literature. Even if PRP-treated cartilage shows a mechanical improvement in animals, this is not directly relevant for guiding human treatment given that there is already a wealth of studies reporting the outcomes of PRP in clinical therapy. This point should be raised and addressed in the discussion.
- IMO is not defined.
Author Response
The reviewer has presented us with important comments and requests, which we believe have improved the quality of our manuscript significantly. Thank you very much. In the following pages, we address each and every comment in detail. Each comment is marked by the reviewer’s original text indexed for convenience of reference, and explains what and how changes were implemented in the new submission. Also, we highlighted all changes in the manuscript in red. We hope the revised manuscript is much closer to the high standard of Molecules.
- Since the study focuses on AFM-based evaluation of articular cartilage, the similarities and differences of rabbit compared to human articular cartilage (nanomechanical properties and other e.g. structural) should be described in more detail. It is stated in the discussion that “Moreover, the size of chondrocytes in human and rabbits articular cartilage do not differ significantly from each other [13].” However, [13] reports a guinea pig model of OA. The reference should reflect a rabbit model of OA.
A: Although animals models don't reflect all aspects of human pathology, they are still used in preclinical trials [12,14,24,50,51]. They play a significant role in understanding OA and the effects of therapeutic interventions. An induced model, causing the initial stages of primary osteoarthritis in rabbits was used in this study. There are some differences between human and rabbit. Human weight-bearing is primarily locked on the knee extension, while rabbit hind limbs are usually kept in a fully flexed position [51]. Also, the cell densities is 1800 and 7500 per mm in humans and rabbits, respectively. Moreover, rabbit stifle joints have different load characteristics and cartilage thickness comparing to humans [51]. Despite those differences, rabbits provide many advantages as they are easy to handle and to house, and cost-effective. Additionally, the rabbit model is the most adequate for this study because it is suitable for the assessment of cartilage repair; rabbits have large enough joints and appropriate size for simple surgical procedures and handling of specimens with a cartilage thickness of 0.25 mm-0.75 mm [52]. The bone mineral density (1.19 g/cm3) is similar to humans at the bone plate (1.17 g/cm3) [53]. Rabbits were used for the assessment of cartilage repair in studies lasting up to 16 weeks [54,55], and animal models of OA, in which articular cartilage damage was induced with intra-articular collagenase injection have proven to be similar to human OA [56].
We added this information in the Discussion section. The reference was corrected [12].
- A brief description should be included in the introduction on possible mechanisms of action for the beneficial characteristics of PRP in treating OA, as well as the current clinical evidence on their efficacy.
A: In the 21st century, orthopaedists’ attention turned to conservative methods of joint disease treatment, indicating that new modalities should focus on multifunctional treatment and solving a set of biotribological [17,18]. Several treatment options were used to stop the progression of inflammation. These include conservative methods, non-steroidal anti-inflammatory drugs, glucocorticosteroids, hyaluronic acid, as well as therapeutic exercise, and assistive devices [18]. Current studies involve the use of platelet-rich plasma therapy (PRP) in the treatment of osteoarthritis because it is easy to prepare and well-tolerated. Some basic, preclinical, and clinical case studies report the ability of platelet-rich plasma to improve musculoskeletal conditions, including osteoarthritis [12,18-25]. Nevertheless, there are still concerns regarding its clinical efficiency, mainly due to the heterogeneity of the used methods and the obtained results, lack of standardization and quantification of the PRP protocols [27-34]. Defining additional parameters related to articular cartilage properties is desirable. We added this information in the Introduction section.
We also added in the Discussion section:
There is a significant gap regarding PRP clinical efficiency. Development of new methods and quantification of the PRP protocols is still under discussion. There are studies in the literature that tested the effect of PRP when used to treat OA in a rabbit model. In [57,58], the authors proved that PRP injection is regenerative in the collagenase-induced knee OA. The PRP treatment group demonstrated a greater extent of cartilage regeneration, as well as higher production of glycosaminoglycans in the extracellular matrix. In [59], the authors reported that the PRP-treated rabbits had the progression of osteoarthritis significantly decreased. Serra et al. [60] also used a rabbit model analyzing the effects of PRP on full-thickness articular cartilage defects. They found that the control group was both microscopically and macroscopically superior concerning defect filling, whereas the PRP group did not show better results compared with placebo.
- How was the dose and frequency of PRP injection in rabbits determined? 4 doses at 7-day intervals is a high frequency of injection that is not clinically viable. Please justify the injection protocol used. Why was 28 days chosen as the final time point? Usually the treatment effects would require monitoring at longer time points. Also, was the first PRP injection performed directly after the OA surgery? If so, this study is not really looking at the effect of PRP on ‘treating’ OA, but rather on preventing the development of OA after an injury stimulus. This point needs to be raised and addressed in the discussion, specifically on the physiological relevance of the study design.
A: The groups 2 and 3 were quarantined for 14 days. Then, group 2 was euthanized by CO inhalation, and group 3 started PRP therapy. The first PRP injection was performed directly after surgery. In another research the OA changes were induced by different doses of 0.5-2.0 mg by the intra-articular injection of [25,58, 61,63]. In our study, the dose was 0.5 mg and frequency of PRP injection was determined as four doses at 7-day intervals. It is in agreement with [64], where authors stated, that the therapeutic effects of plasma therapy are based on the release from platelets – flat, nuclear-free cells that circulate in the bloodstream with a lifespan of 7-10 days.
Usually, the treatment effects require monitoring at longer time period. Therefore, we speculate that 0.5 mg PRP injection is sufficient to evaluate morphological changes with AFM shortly after PRP injection. We added the text in the Discussion section.
- There are other studies in the literature that tested the effect of PRP when used to treat OA in a rabbit model, some of which included AFM as one of the outcome measures. These studies should be cited in the discussion and the results of the current study should be compared. Although mentioned in the discussion, the novelty of the current study should be more clearly stated in the context of the existing literature.
A: Radiography, currently the gold standard for OA imaging and other image modalities have limitations in the area of providing information about the mechanical properties of cartilage altered by degenerative joint disease. The atomic force microscopy (AFM) was widely utilized in the detection of pathological and the biomechanical properties of biological target structures on microscopic scale conditions [65-70]. In our study, AFM was used to test the mechanical properties and the submicron surface morphology of rabbits’ hyaline cartilage. Pronounced heterogeneity of the biological surfaces’ topography renders the utilization of AFM method quite challenging. Thus, we proposed a measurement protocol for the AFM method for determining the nanomechanical properties of articular cartilage. We pre-selected the silicon probes with a single radius of the needlepoint using a scanning electron microscope (±1 nm) and with cantilever constant (±0.01 N/m) by testing samples with the previously known Young’s modulus (NaCl crystal with E=37 GPa) to reduce error. The AFM indentation mode can be realized in a few ways depending on the set controlled parameter. We used the set-point as a controlled parameter. The set-point corresponds to relative deflection of the probe cantilever in the static mode of indentation. AFM moves the sample step-by-step towards the probe in such way that set-point changes in the preset range with equal steps between the set-point values. Measured cantilever deflection is a function of set-point. The dependence of the cantilever deviation on the set-point parameter is established, based on the measurement results. The cantilever deflection can be directly converted into force if the mechanical properties of the cantilever are known. Chandran et al. [71] stated that the indentation depth of at least 0.6 μm is required to obtain values of matrix elasticity. It was observed that each 60 μm change in indenter location could result in a 20-fold variation of the measurement [72]. Our previous research carried out to 450 nm in depth [49] shows that the surface is more suitable to diagnose, so in this study, we measured the mechanical properties of cartilage surface to 150 nm in depth. However, the statistical significance of the results was observed below 75 nm. Past studies of articular cartilage have assumed a wide range of Poisson’s ratios [69,70]. In this study [49], the local Poisson’s ratio was assumed to be 0.5 for articular cartilage based on previous experimental reports. The changes in articular cartilage structure result in deterioration of the mechanical strength [66]. The roughness parameters of hyaline cartilage surface: the absolute value (Ra), the maximum peak height (Rp), and the mean spacing between local peaks (S) were measured for three groups: control; Induced Model of Osteoarthritis; and the group with PRP treatment. Our results show that PRP injection into the knee under the patella can improve the mechanical properties and the submicron surface morphology of rabbits’ hyaline cartilage: Ra and Rp increased, while S decreased comparing to healthy rabbits cartilage (p<0.05). Moreover, Ra, Rp and S decreased comparing to group 2. The extent of mechanical compliance can be expressed as Young's modulus, which describes the elastic properties of a material [67,68]. In our study, the indentation clearly revealed a decrease in the Young’s modulus of the upper layers of hyaline cartilage (up to 100 nm). It increased by 9% as a result of PRP therapy, which means that the mechanical properties of cartilage partially recovered. We added the text in the Discussion section
- The clinical relevance of this study should be better defined. The main limitation of this study is that only AFM characterisation of the cartilage has been performed, which does not give much useful information in the context of the existing literature. Even if PRP-treated cartilage shows a mechanical improvement in animals, this is not directly relevant for guiding human treatment given that there is already a wealth of studies reporting the outcomes of PRP in clinical therapy. This point should be raised and addressed in the discussion.
A: Although the roughness measurements are routine in the presented field, the novelty of our study lies in using surface nano-roughness and microhardness of subsurface layers in describing changes in articular cartilage properties during development of OA. These parameters can be treated as essential indicators of the pathologic progression of OA diseases. In addition, the elastic modulus determined by AFM indentation experiments, can be utilized as a progressive disease marker in preventing the development of OA.
Although animals models don't reflect all aspects of human pathology, they are still used in preclinical trials [12,14,24,50,51]. They play a significant role in understanding OA and the effects of therapeutic interventions. An induced model, causing the initial stages of primary osteoarthritis in rabbits was used in this study. There are some differences between human and rabbit. Human weight-bearing is primarily locked on the knee extension, while rabbit hind limbs are usually kept in a fully flexed position [51]. Also, the cell densities is 1800 and 7500 per mm in humans and rabbits, respectively. Moreover, rabbit stifle joints have different load characteristics and cartilage thickness comparing to humans [51]. Despite those differences, rabbits provide many advantages as they are easy to handle and to house, and cost-effective. Additionally, the rabbit model is the most adequate for this study because it is suitable for the assessment of cartilage repair; rabbits have large enough joints and appropriate size for simple surgical procedures and handling of specimens with a cartilage thickness of 0.25 mm-0.75 mm [52]. The bone mineral density (1.19 g/cm3) is similar to humans at the bone plate (1.17 g/cm3) [53]. Rabbits were used for the assessment of cartilage repair in studies lasting up to 16 weeks [54,55], and animal models of OA, in which articular cartilage damage was induced with intra-articular collagenase injection have proven to be similar to human OA [56].
To the best of the authors’ knowledge, this is the first study to assess the response to treatment consisting of intra-articular PRP injection on the mechanical properties and the submicron surface morphology of hyaline cartilage. The limitations of this study included the sample size and lack of therapeutic effects of PRP in severe knee OA during long time. In the future, we consider using a liquid cell for AFM measurements, because it eliminates surface tension and makes it possible to perform measurements immediately after obtaining the specimens. We added the text in the Discussion section
We also added in the Methods and Results section:
2.3. Histologic parameters
Rabbits articular cartilage was harvested after euthanasia from the femoral condyles, yielding ∼5 mm × 5 mm pieces, ∼2 mm thick. All samples were fixed in 10% neutral buffered formalin for at least 48 h. For histopathological examination, tissue samples were washed with 10% phosphate buffered saline solution, dehydrated in alcohol and embedded in paraffin wax [13]. The degree of cartilage degradation was assessed using the scoring system modified by Mankin et al. [48]. A pathologist evaluated the structural change of articular cartilage as histological evidence of cartilage degeneration.
3.1. Histologic findings according to the non-treated and PRP-injected knees
The difference of histologic scores in group 3 was higher (3.3 (1.4)) than those in group 2 (2.1 (1.8)), p<0.05. Histologic examination showed nearly normal cartilage surfaces with normal cellularity in the transitional and radial zones in group 3 with PRP injection.
- IMO is not defined.
A: Done (in figure 3).

Round 2
Reviewer 3 Report
The manuscript has been significantly improved. I have no more comments to add.